# Prevalence and impact of multidrug-resistant bacteria in solid cancer patients with bloodstream infection: a 25-year trend analysis

Carlos Lopera,[1] Patricia Monzó,[1] Tommaso Francesco Aiello,[1] Mariana Chumbita,[1] Olivier Peyrony,[2] Antonio Gallardo-Pizarro,[1] Cristina Pitart,[3] Guillermo Cuervo,[1] Laura Morata,[1] Marta Bodro,[1] Sabina Herrera,[1] Ana Del Río,[1] José Antonio Martínez,[1] Alex Soriano,[1,4] Pedro Puerta-Alcalde,[1,4] Carolina Garcia-Vidal[1,4]

**ABSTRACT** The study aimed to describe the epidemiology of multidrug-resistant (MDR) bacteria among solid cancer (SC) patients with bloodstream infections (BSIs), evaluating inappropriate empiric antibiotic treatment (IEAT) use and mortality trends over a 25-year period. All BSI occurrences in adult SC patients at a university hospital were analyzed across five distinct five-year intervals. MDR bacteria were classified as extended-spectrum beta-lactamases (ESBL)-producing and/or Carbapenem-resistant Enterobacterales, non-fermenting Gram-negative bacilli (GNB) resistant to at least three antibiotic classes, methicillin-resistant *Staphylococcus aureus* (MRSA), and Vancomycin-resistant *Enterococci*. A multivariate regression model identified the risk factors for MDR BSI. Of 6,117 BSI episodes, Gram-negative bacilli (GNB) constituted 60.4% (3,695/6,117), being the most common are *Escherichia coli* with 26.8% (1,637/6,117), *Klebsiella* spp. with 12.4% (760/6,117), and *Pseudomonas aeruginosa* with 8.6% (525/6,117). MDR-GNB accounted for 644 episodes (84.8% of MDR or 644/759), predominantly ESBL-producing strains (71.1% or 540/759), which escalated significantly over time. IEAT was administered in 24.8% of episodes, mainly in MDR BSI, and was associated with higher mortality (22.9% vs. 14%, $P < 0.001$). Independent factors for MDR BSI were prior antibiotic use [odds ratio (OR) 2.93, confidence interval (CI) 2.34–3.67], BSI during antibiotic treatment (OR 1.46, CI 1.18–1.81), biliary (OR 1.84, CI 1.34–2.52) or urinary source (OR 1.86, CI 1.43–2.43), admission period (OR) 1.28, CI 1.18–1.38, and community-acquired infection (OR 0.57, CI 0.39–0.82). The study showed an increase in MDR-GNB among SC patients with BSI. A quarter received IEAT, which was linked to increased mortality. Improving risk assessment for MDR infections and the judicious prescription of empiric antibiotics are crucial for better outcomes.

**IMPORTANCE** Multidrug-resistant (MDR) bacteria pose a global public health threat as they are more challenging to treat, and they are on the rise. Solid cancer patients are often immunocompromised due to their disease and cancer treatments, making them more susceptible to infections. Understanding the changes and trends in bloodstream infections in solid cancer patients is crucial, to help physicians make informed decisions about appropriate antibiotic therapies, manage infections in this vulnerable population, and prevent infection. Solid cancer patients often require intensive and prolonged treatments, including surgery, chemotherapy, and radiation therapy. Infections can complicate these treatments, leading to treatment delays, increased healthcare costs, and poorer patient outcomes. Investigating new strategies to combat MDR infections and researching novel antibiotics in these patients is of paramount importance to avoid these negative impacts.

Address correspondence to Carolina Garcia-Vidal, cgarciav@clinic.cat.

Pedro Puerta-Alcalde and Carolina Garcia-Vidal contributed equally to this article.

C.G.-V. has received honoraria for talks on behalf of Gilead Sciences, MSD, Novartis, Pfizer, Janssen, and Lilly, as well as a grant from Gilead Sciences, Pfizer, and MSD. P.P.-A. has received honoraria for talks on behalf of Merck Sharp & Dohme, Gilead, Lilly, ViiV Healthcare, and Gilead Sciences. A.S. has received honoraria for talks on behalf of Merck Sharp & Dohme, Pfizer, Novartis, Angelini, Menarini, and Gilead Sciences, as well as grant support from Pfizer and Gilead Sciences. O.P. has received honoraria for talks on behalf of MSD, Qiagen, and expertise for Sanofi.

See the funding table on p. 10.

KEYWORDS   bloodstream infections, solid cancer, inappropriate empirical antibiotic therapy, Gram-negative bacilli, multidrug-resistant

In recent decades, oncology has undergone a major revolution with the introduction of increasingly personalized treatments. Such therapies have resulted in a significant decrease in toxicities and an optimal improvement in the survival of patients with different types of solid cancer (SC) (1). However, infections remain a significant complication for these patients because of their high morbidity and mortality (2, 3).

Bloodstream infections (BSIs) are one of the most severe infectious diseases for these patients (4–7). In recent years, the rapid emergence of multidrug-resistant microorganisms (8–10) has posed a major challenge with respect to the decisions about empiric antibiotic treatments (11, 12). This holds especially true because these patients have a closer relationship with the healthcare environment. This exposure often leads to a frequent contact with multidrug-resistant (MDR) bacteria and extensive use of antibiotics that contributes to the selective pressure promoting the development of new resistant strains (8–10). Yet, knowledge on the current epidemiology of BSI in patients with SC is scarce.

In this study, we aimed to describe the changing epidemiology focusing on the microbial distribution, clinical features, and incidence of MDR bacteria in our 25-year cohort of patients with SC and BSI. Moreover, we detailed the percentage of patients who received inappropriate empiric antibiotic treatment (IEAT), as well as the 30-day mortality trend.

## MATERIALS AND METHODS

### Setting, data collection and study population

We performed this study at the Hospital Clinic in Barcelona (Spain), a 700-bed university center that provides broad and specialized medical, surgical, and intensive care for an urban population of 500,000 people. Our center annually treats more than 2,000 patients with SC, with a steady increase of such a population over time.

Since 1995, our institution has overseen a blood culture surveillance program that identifies all patients with bloodstream infections. Data on epidemiology, clinical features, microbiology, treatment, and outcomes have been prospectively recorded. We analyzed all consecutive episodes of BSI occurring in adults with SC between 1995 and 2019. For this study, we divided this 25-year period into five different five-year time spans. The ethics committee board of our institution approved this study.

### Definitions

Definitions for comorbidities, prognosis of underlying disease, and source of infections, have been previously provided (2, 4). The selection of antibiotic treatment for the patients was conducted by the attending physicians without any influence from the article's investigators. Treatment protocols take into account a syndromic antibiotic approach (such as pneumonia, urinary tract infection, etc.), the origin of the infection (community-acquired or nosocomial), and the local ecology. For those patients with neutropenia, the protocols align with national guidelines (13).

Neutropenia was defined as an absolute neutrophil count ≤500 cells/mm. Appropriate empiric therapy was considered when the patient received at least one *in vitro* active empiric antimicrobial agent, and the dosage and route of administration were in accordance with current medical standards. Mortality was defined as death occurring within 30 days of bacteremia onset. Death was considered related to the BSI if it occurred before the resolution of symptoms or signs, or within 7 days of the onset of bacteremia, and there was no other explanation. The following Gram-negative bacilli (GNB) were classified as MDR: (1) extended-spectrum beta-lactamase (ESBL)-producing Enterobacterales; (2) Carbapenem-resistant Enterobacterales; and (3) non-fermenting GNB that are

resistant to at least three classes of these antibiotics, namely carbapenems, ureidope-nicillins, cephalosporins (ceftazidime and cefepime), monobactams, aminoglycosides, and fluoroquinolones. Extensively drug-resistant (XDR)-GNB were those non-susceptible to ≥1 agent in all but ≤2 categories (14). The following Gram-positive cocci (GPC) were classified as MDR: (1) methicillin-resistant *Staphylococcus aureus* (MRSA) and (2) vancomycin-resistant enterococci (VRE).

## Microbiologic methods

Blood samples were processed using the BACTEC 9240 system or BACTEC FX system (Becton Dickinson Microbiology Systems), with a 5-day incubation period. Isolates were identified by standard techniques. A single positive culture was considered as significant if the microbe was clinically relevant or isolated in blood cultures. Of note, for BSI, due to coagulase-negative staphylococci (CoNS) whereby no catheter tip culture was available, at least two sets of positive blood cultures from different venipuncture sites were required to be considered as significant. Antimicrobial susceptibility testing was performed with either an automated microdilution system (MicroScan WalkAway, Beckman Coulter, West Sacramento, CA, USA; Phoenix system, Becton Dickinson, Franklin Lakes, NJ, USA), a semiautomated method (Sensititre, Thermo Fisher Scientific), or gradient strips (Etest; AB BIODISK, Solna, Sweden/bioMérieux, Marcy-l'Étoile, France). ESBLs were suspected by minimum inhibitory concentration (MIC) results and confirmed by double-disc synergy testing (15). Carbapenemase-producing Enterobacter-ales were phenotypically detected by the modified carbapenem inactivation method (mCIM) (16), in combination with the NG-Test CARBA 5 lateral flow immunoassay (NG Biotech, France) to detect the five most prevalent carbapenemases [*Klebsiella pneumo-niae* carbapenemase (KPC), OXA-48-like, Verona integron-encoded metallo-β-lactamase (VIM), imipenemase (IMP), and New Delhi metallo-β-lactamase (NDM)] (17). We used the current Clinical and Laboratory Standards Institute (CLSI) (from 1995 to 2009) and European Committee on Antimicrobial Susceptibility Testing (EUCAST) breakpoints (from 2010 to 2019) for each year to define the susceptibility or resistance to these antimicrobial agents; intermediate susceptibility was considered as resistance.

## Statistical analysis

Categorical variables were detailed as counts and percentages, whereas continuous variables were described as either means and standard deviations (SDs) or medians and interquartile ranges (IQRs). The Pearson's $\chi^2$ test and either the Mann-Whitney U test or the Student's *t*-test according to the variable distribution were employed to compare the distribution of categorical and continuous variables, respectively. The Ordinary Least Squares (OLS) linear regression model was applied to analyze the temporal trend. We calculated the annual percentage change (APC) and its 95% confidence intervals (95% CI). A trend was identified as increasing with a positive regression coefficient or decreasing with a negative one when $P < 0.05$. Additionally, diagnostic tests were conducted to assess the assumptions for autocorrelation and the validity of the linear regression model. Chi-square for trends was used to compare the different time spans. A multivariate regression model (step-forward procedure) was used to identify independent risk factors for MDR BSI. We constructed a regression model with the inclusion of the MDR BSIs as the dependent variable. For the independent variables, we chose those parameters that showed predictive value using univariate analysis ($P < 0.05$). The goodness-of-fit of the multivariate model was assessed with the Hosmer-Lemeshow test and the area under the receiver operating characteristic curve. Our research utilized IBM SPSS Statistics for Windows (Version 25.0, Armonk, NY: IBM Corp.) and the Python programming language for data analysis.

## RESULTS

### Cohort and BSI characteristics

We documented a total of 5,896 patients with SC and BSI. The median age of the patients was 68 (IQR 59–76) years, and 3,835 (65.0%) were males. The most common SC was colorectal cancer (19.5%), followed by hepatobiliary cancers (14.3%), and urothelial cancer (13.5%). Only 333 (5.7%) patients were neutropenic and 947 (16.5%) had previously received corticosteroids. The urinary source of BSI was the most common (23.2%), followed by catheter-related BSI (20.8%) and biliary-tract BSI (10.9%). Table 1 shows the changes in patients and BSI characteristic over the study period.

### Causative microorganisms and antibiotic resistance

A total of 6,117 different pathogens were isolated in the 5,896 patients with SC and BSI. Overall, 36.3% of BSI were due to GPC and 60.4%, to Gram-negative bacilli (GNB). The most common GNB were *Escherichia coli* (1,637, 26.8%), *Klebsiella* spp. (760, 12.4%),

**TABLE 1**  Main characteristics of patients, source of BSI and outcomes over the five-year time spans[b,c]

| Number of patients with SC[a] (%) | 1995–1999 | 2000–2004 | 2005–2009 | 2010–2014 | 2015–2019 | Overall | P-value for |
| --- | --- | --- | --- | --- | --- | --- | --- |
| | 874 (14.8) | 1147 (19.5) | 1302 (22.1) | 1254 (21.3) | 1319 (22.4) | 5896 (100) | trend |
| Male sex | 524 (60.0) | 730 (63.6) | 852 (65.4) | 853 (68.0) | 876 (66.4) | 3835 (65.0) | <0.001 |
| Median (IQR) age, in years | 68 (58–76) | 68 (58–77) | 68 (52–84) | 68 (60–76) | 68 (59–77) | 68 (60–77) | NS |
| Baseline disease | | | | | | | |
| Diabetes mellitus | 137 (15.7) | 177 (15.4) | 194 (14.9) | 249 (19.9) | 247 (18.7) | 1004 (17.0) | 0.002 |
| Chronic hepatopathy | 106 (12.1) | 146 (12.7) | 116 (8.9) | 131 (10.4) | 97 (7.4) | 596 (10.1) | <0.001 |
| Chronic heart disease | 58 (6.6) | 104 (9.1) | 109 (8.4) | 138 (11.0) | 123 (9.3) | 532 (9.0) | 0.014 |
| Chronic lung disease | 65 (7.4) | 78 (6.8) | 85 (6.5) | 112 (8.9) | 91 (6.9) | 431 (7.3) | NS |
| Chronic kidney disease | 47 (5.4) | 31 (2.7) | 67 (5.1) | 82 (6.5) | 124 (9.4) | 351 (6.0) | <0.001 |
| Type of SC[a] | | | | | | | |
| Colorectal cancer | 121 (13.8) | 216 (18.8) | 294 (22.6) | 284 (22.6) | 235 (17.8) | 1150 (19.5) | 0.015 |
| Hepatobiliary cancers | 134 (15.3) | 166 (14.5) | 169 (13) | 175 (14) | 198 (15) | 842 (14.3) | NS |
| Urothelial cancer | 83 (9.5) | 125 (10.9) | 195 (15.0) | 184 (14.7) | 210 (15.9) | 797 (13.5) | <0.001 |
| Pancreatic cancer | 62 (7.1) | 65 (5.7) | 99 (7.6) | 119 (9.5) | 150 (11.4) | 495 (8.4) | <0.001 |
| Breast cancer | 91 (10.4) | 92 (8.0) | 100 (7.7) | 73 (5.8) | 91 (6.9) | 447 (7.6) | 0.002 |
| Lung cancer | 61 (7.0) | 106 (9.2) | 95 (7.3) | 83 (6.6) | 95 (7.2) | 440 (7.5) | NS |
| Prostate cancer | 48 (5.5) | 90 (7.8) | 90 (6.9) | 94 (7.5) | 106 (8.0) | 428 (7.3) | NS |
| Gastro-oesophageal cancer | 82 (9.4) | 51 (4.4) | 84 (6.5) | 61 (4.9) | 70 (5.3) | 348 (5.9) | 0.003 |
| Gynecological cancer | 43 (4.9) | 62 (5.4) | 65 (5) | 59 (4.7) | 84 (6.4) | 313 (5.3) | NS |
| Other[a] | 185 (21.2) | 222 (19.4) | 151 (11.6) | 163 (13) | 192 (14.6) | 913 (15.5) | <0.001 |
| Clinical conditions | | | | | | | |
| Neutropenia | 78 (9) | 64 (5.6) | 59 (4.6) | 66 (5.4) | 66 (5.1) | 333 (5.7) | 0.002 |
| Prior corticosteroids within the last 3 months. | 141 (16.2) | 237 (20.8) | 154 (12.6) | 201 (16.9) | 214 (16.5) | 947 (16.5) | NS |
| Central venous catheter | 339 (78.8) | 457 (58.1) | 599 (55.1) | 515 (76.8) | 512 (74.9) | 2422 (66.2) | <0.001 |
| Prior antibiotic therapy (1 month) | 319 (36.5) | 372 (32.4) | 488 (37.5) | 565 (45.1) | 645 (48.9) | 2389 (40.5) | <0.001 |
| Antibiotic therapy at BSI onset | 288 (33) | 276 (24.1) | 308 (23.7) | 344 (27.4) | 315 (23.9) | 1531 (26) | 0.003 |
| Most frequent source of BSI | | | | | | | |
| Urinary | 146 (16.7) | 240 (20.9) | 304 (23.3) | 317 (25.3) | 362 (27.4) | 1369 (23.2) | <0.001 |
| Catheter | 211 (24.1) | 218 (19.0) | 309 (23.7) | 287 (22.9) | 200 (15.2) | 1225 (20.8) | <0.001 |
| Endogenous/Unknown | 176 (20.1) | 270 (23.5) | 281 (21.6) | 207 (16.5) | 231 (17.5) | 1165 (19.8) | <0.001 |
| Biliary tract | 87 (10.0) | 104 (9.1) | 135 (10.4) | 127 (10.1) | 187 (14.2) | 640 (10.9) | <0.001 |
| Intra-abdominal | 71 (8.1) | 99 (8.6) | 84 (6.5) | 97 (7.7) | 113 (8.6) | 464 (7.9) | NS |
| Pulmonary source | 77 (8.8) | 90 (7.8) | 71 (5.5) | 55 (4.4) | 64 (4.9) | 357 (6.1) | <0.001 |
| Septic shock | 119 (13.7) | 189 (16.5) | 166 (12.8) | 191 (15.3) | 195 (15.0) | 860 (14.7) | NS |

[a]Including oropharyngeal (208), central nervous system (120), melanoma (111), sarcoma (84), and other solid cancers (289).
[b]Abbreviations: SC, solid cancer; BSI, bloodstream infection.
[c]All percentages are according to the number of patients with SC in the defined period.

and *Pseudomonas aeruginosa* (525, 8.6%). Coagulase-negative staphylococci (CoNS) (701, 11.5%), *Enterococcus* spp. (584, 9.5%), *S. aureus* (470, 7.7%), and *Streptococcus* spp. (464, 7.6%) were the most frequent GPC. Additionally, there were 203 (3.3%) episodes of candidemia and 617 (10.1%) polymicrobial episodes. Table 2 details causative microorganisms and MDR isolates over time.

Over the study period, there were 759 (12.4%) BSI episodes caused by MDR isolates, 115 (2.0%) by MDR-GPC, and 644 (10.9%) by MDR-GNB. A significant increase in the rates of multidrug-resistant organisms (MDROs) was observed over the study period (*P* <

**TABLE 2** Changes over time in BSI epidemiology in patients with solid cancer[e]

| Number of BSI (%) per period | 1995–1999 *n* = 900 (14.7) | 2000–2004 *n* = 1183 (19.3) | 2005–2009 *n* = 1352 (22.1) | 2010–2014 *n* = 1325 (21.7) | 2015–2019 *n* = 1357 (22.2) | Overall *n* = 6117 (100) | *P*-value for trend |
|---|---|---|---|---|---|---|---|
| Gram-positive cocci | 470/900 (52.2) | 503/1183 (42.5) | 463/1352 (34.2) | 394/1325 (29.8) | 389/1357 (28.7) | 2219/6117 (36.3) | <0.001 |
| Coagulase-negative staphylococci | 184/900 (20.4) | 193/1183 (16.3) | 151/1352 (11.2) | 89/1325 (6.7) | 84/1357 (6.2) | 701/6117 (11.5) | <0.001 |
| *Enterococcus* spp. | 83/900 (9.2) | 81/1183 (6.8) | 123/1325 (9.1) | 149/1325 (11.2) | 148/1357 (10.9) | 584/6117 (9.5) | 0.002 |
| *E. faecali*/Enterococcus spp. | 66/83 (79.5) | 61/81 (75.3) | 88/123 (71.5) | 89/149 (59.7) | 68/148 (45.9) | 372/584 (63.7) | NS |
| *E. faecium*/Enterococcus spp. | 12/83 (14.4) | 14/81 (17.3) | 34/123 (27.6) | 55/149 (36.9) | 78/148 (49.4) | 193/584 (33.0) | <0.001 |
| Vancomycin-resistant *Enterococcus*/*Enterococcus* spp. | 2/83 (2.4) | 3/81 (3.7) | 3/123 (2.4) | 5/149 (3.4) | 2/148 (1.4) | 15/584 (2.6) | NS |
| *S. aureus* | 95/900 (10.6) | 119/1183 (10.1) | 103/1352 (7.6) | 78/1325 (5.9) | 75/1357 (5.5) | 470/6117 (7.7) | <0.001 |
| MRSA[a]/*S. aureus* | 27/95 (28.4) | 24/119 (20.2) | 24/103 (23.3) | 12/78 (15.4) | 13/75 (17.3) | 100/470 (21.3) | <0.001 |
| *Streptococcus* spp. | 108/900 (12.0) | 110/1183 (9.3) | 86/1352 (6.4) | 78/1325 (5.9) | 82/1357 (6.0) | 464/6117 (7.6) | <0.001 |
| Gram-negative bacilli (GNB) | 400/900 (44.4) | 653/1183 (55.2) | 833/1352 (61.6) | 878/1325 (66.3) | 931/1357 (68.6) | 3695/6117 (60.4) | <0.001 |
| *E. coli* | 209/900 (23.2) | 326/1183 (27.6) | 355/1352 (26.3) | 367/1325 (27.7) | 380/1357 (28.0) | 1637/6117 (26.8) | 0.024 |
| ESBL[a] *E. coli*/*E. coli* | 4/209 (1.9) | 22/326 (6.7) | 57/355 (16.1) | 71/367 (19.3) | 79/380 (20.8) | 233/1637 (14.2) | <0.001 |
| Carbapenem-resistant *E. coli*/*E. coli* | 0 | 0 | 3/355 (0.8) | 1/367 (0.3) | 5/380 (1.3) | 9/1637 (0.5) | 0.021 |
| *Klebsiella* spp. | 46/900 (5.1) | 70/1183 (5.9) | 180/1352 (13.3) | 198/1325 (14.9) | 266/1357 (19.6) | 760/6117 (12.4) | <0.001 |
| ESBL[a] *Klebsiella* spp/*Klebsiella* spp | 1/46 (2.2) | 7/70 (10) | 38/180 (21.1) | 50/198 (25.3) | 95/266 (35.7) | 191/760 (25.1) | <0.001 |
| Carbapenem-resistant *Klebsiella* spp/*Klebsiella* spp. | 0 | 0 | 1/180 (0.6) | 11/198 (5.6) | 24/266 (9.0) | 36/760 (4.7) | <0.001 |
| *P. aeruginosa* | 51/900 (5.7) | 100/1183 (8.5) | 119/1352 (8.8) | 142/1325 (10.7) | 113/1357 (8.3) | 525/6117 (8.6) | 0.010 |
| MDR *P. aeruginosa*[b]/*P. aeruginosa* | 6/51 (11.8) | 8/100 (8.0) | 24/119 (20.2) | 25/142 (17.6) | 11/113 (9.7) | 74/525 (14.1) | NS |
| XDR *P. aeruginosa*[c]/*P. aeruginosa* | 4/51 (7.8) | 4/100 (4.0) | 19/119 (16.0) | 24/142 (16.9) | 11/113 (9.7) | 62/525 (11.8) | 0.028 |
| *Enterobacter* spp. | 19/900 (2.1) | 42/1183 (3.6) | 75/1352 (5.5) | 70/1325 (5.3) | 62/1357 (4.6) | 268/6117 (4.4) | 0.002 |
| ESBL[a] *Enterobacter* spp./*Enterobacter* spp. | 5/19 (26.3) | 8/42 (19.0) | 29/75 (38.7) | 16/70 (22.9) | 23/62 (37.1) | 81/268 (30.2) | 0.013 |
| Other Enterobacterales | 42/900 (4.7) | 74/1183 (6.3) | 63/1352 (4.7) | 64/1325 (4.8) | 71/1357 (5.2) | 314/6117 (5.1) | NS |
| ESBL[a] producing other Enterobacterales/other Enterobacterales | 5/42 (11.9) | 8/74 (10.8) | 5/63 (7.9) | 8/64 (12.5) | 9/71 (12.7) | 35/314 (11.1) | NS |
| Carbapenem-resistant *Klebsiella* spp./*Klebsiella* spp. | 0 | 0 | 1/180 (0.6) | 11/198 (5.6) | 24/266 (9.0) | 36/760 (4.7) | <0.001 |
| *S. maltophilia* | 7/900 (0.8) | 8/1183 (0.7) | 7/1352 (0.5) | 9/1325 (0.7) | 16/1357 (1.2) | 47/6117 (0.8) | NS |
| Polymicrobial[d] | 93/900 (10.3) | 113/1183 (9.6) | 109/1352 (8.1) | 160/1325 (12.1) | 142/1357 (10.5) | 617/6117 (10.1) | NS |
| Candidemia | 30/900 (3.3) | 27/1183 (2.3) | 56/1352 (4.1) | 53/1325 (4.0) | 37/1357 (2.7) | 203/6117 (3.3) | NS |
| MDR-GPC[a] | 29/470 (6.2) | 27/503 (5.4) | 27/463 (5.8) | 17/394 (4.3) | 15/389 (3.9) | 115/2219 (5.2) | <0.001 |
| MDR-GNB[a] | 29/400 (7.3) | 61/653 (9.3) | 157/833 (18.8) | 172/878 (19.6) | 225/931 (24.2) | 644/3695 (17.4) | <0.001 |
| Any MDR[a] | 58/900 (6.4) | 88/1183 (7.4) | 184/1352 (13.6) | 189/1325 (14.3) | 240/1357 (17.7) | 759/6117 (12.4) | <0.001 |

[a]Abbreviations: MRSA, methicillin-resistant *Staphylococcus aureus*; ESBL, extended-spectrum beta-lactamase; MDR, multiple drug-resistant; XDR, extensively drug-resistant; GPC, Gram-positive cocci; GNB, Gram-negative bacilli.
[b]MDR *P. aeruginosa* is defined as non-susceptible to at least one agent in three or more antimicrobial categories.
[c]XDR *P. aeruginosa* is defined as non-susceptibility to at least one agent in all but two or fewer antimicrobial categories.
[d]Polimicrobial infections: 263 were GPC and GNB infections, 69 were caused by two or more GPC, four were polymicrobial GPC (including at least two GPC), and at least one GNB at the same time, 12 episodes were caused by candidemia and GPC, 15 episodes were caused by candidemia and GNB, and four episodes of candidemia included at least two different species of Candida, there were also 250 episodes of polymicrobial GNB infections.
[e]Antimicrobial categories and agents used to define MDR and XDR *P. aeruginosa* include aminoglycosides, antipseudomonal carbapenems, antipseudomonal cephalosporins, antipseudomonal fluoroquinolones, antipseudomonal penicillins + β-lactamase inhibitors, monobactams, phosphonic acids, and polymyxins.

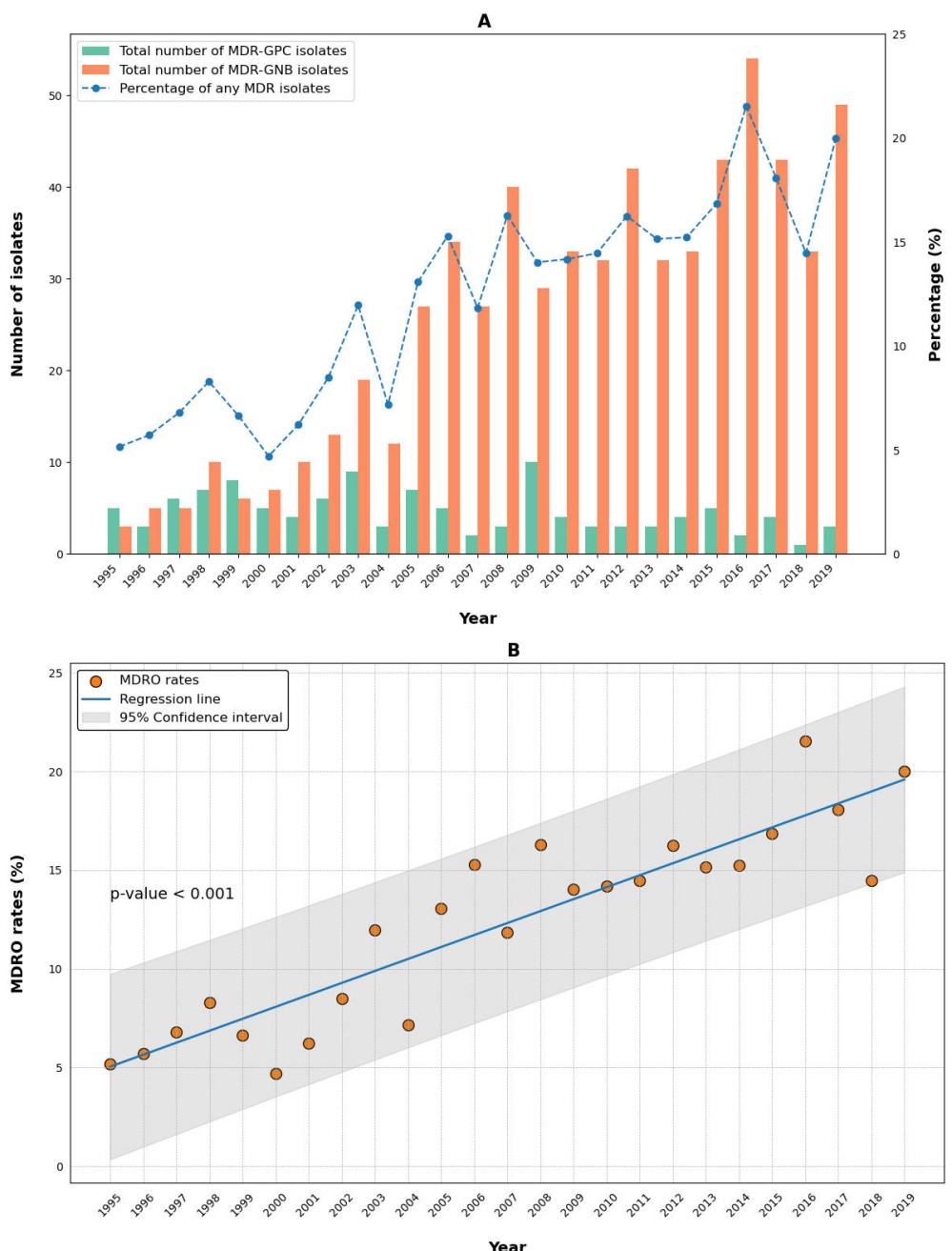

**FIG 1** Temporal trends in annual incidence rates of MDRO in bloodstream infections over 25 years. MDRO, multidrug-resistant organism; MDR-GPC, multiple drug-resistant Gram-positive cocci; MDR-GNB, multiple drug-resistant Gram-negative bacilli.

0.001), with an APC of 0.57% (95% CI 0.49; 0.69) (Fig. 1). MDR-GPC significantly decreased over time ($P$ < 0.001) due to a decline in the prevalence of BSI caused by MRSA ($P$ < 0.001). Conversely, BSI caused by MDR-GNB increased ($P$ < 0.001), mainly secondary to the rise in ESBL-producing *Klebsiella* spp. and ESBL-producing *E. coli* ($P$ < 0.001 for both).

Of the 644 MDR-GNB BSI documented, the most common isolates were 233 (3.8% of total BSIs) ESBL-producing *E. coli*, 191 (3.1%) ESBL-producing *Klebsiella* spp., and 74 (1.2%) MDR *P. aeruginosa*. Fig. 2 and 3 detail the changes in BSI antimicrobial resistance caused by the main Enterobacterales and *P. aeruginosa* over time, respectively.

In the last time span of years (2015–2019), 20.8% of *E. coli* isolates and 35.7% of *Klebsiella* spp. were ESBL producers and 1.3% and 9.0%, respectively, were

carbapenem-resistant. Regarding *P. aeruginosa*, 21.2% of all isolates were resistant to one of the most common antipseudomonal beta-lactams (ceftazidime, piperacillin-tazobactam and/or meropenem); 14.1% were MDR strains and 11.8% met criteria for XDR.

## Appropriateness of empiric antibiotic therapy and mortality

Over the study period, 1,520 (24.8%) BSI episodes received IEAT. In Enterobacterales, IEAT was 15.2% and more frequent among ESBL (42% vs 15.2%, $P < 0.001$) and carbapenem-resistant isolates (51.1% vs 15.2%, $P < 0.001$). In *P. aeruginosa*, 31.3% episodes received IEAT, and it was significantly more frequent in MDR (58.1% vs 31.4%, $P < 0.001$) and XDR (62.9% vs 31.4%) *P. aeruginosa*. Table 3 summarises the changes over time on IEAT rates.

Thirty-day mortality was 16.3% and remained stable throughout the study period ($P = 0.355$). Mortality was significantly higher in patients receiving IEAT (22.9% vs 14%, $P < 0.001$).

## Risk factors for BSI caused by MDR strains

In the multivariate analysis, the independent risk factors for a MDR BSI in our cohort of patients were: prior antibiotic use (OR 2.93, CI 2.34–3.67), BSI occurring during antibiotic treatment (OR 1.46, CI 1.18–1.81), biliary source of BSI (OR 1.84, CI 1.34–2.52), urinary source of BSI (OR 1.86, CI 1.43–2.43), and period of admission (OR 1.28, CI 1.18–1.38). Community-acquired infection was related with lower risk of MDR BSI (OR 0.57, CI 0.39–0.82). The goodness of fit of the multivariate model was assessed by the Hosmer-Lemeshow test (0.644), and the discriminatory power of the score, as evaluated by the area under the receiver operating characteristic curve, was 0.72 (95% CI, 0.70–0.74), demonstrating a good ability to predict MDR BSI.

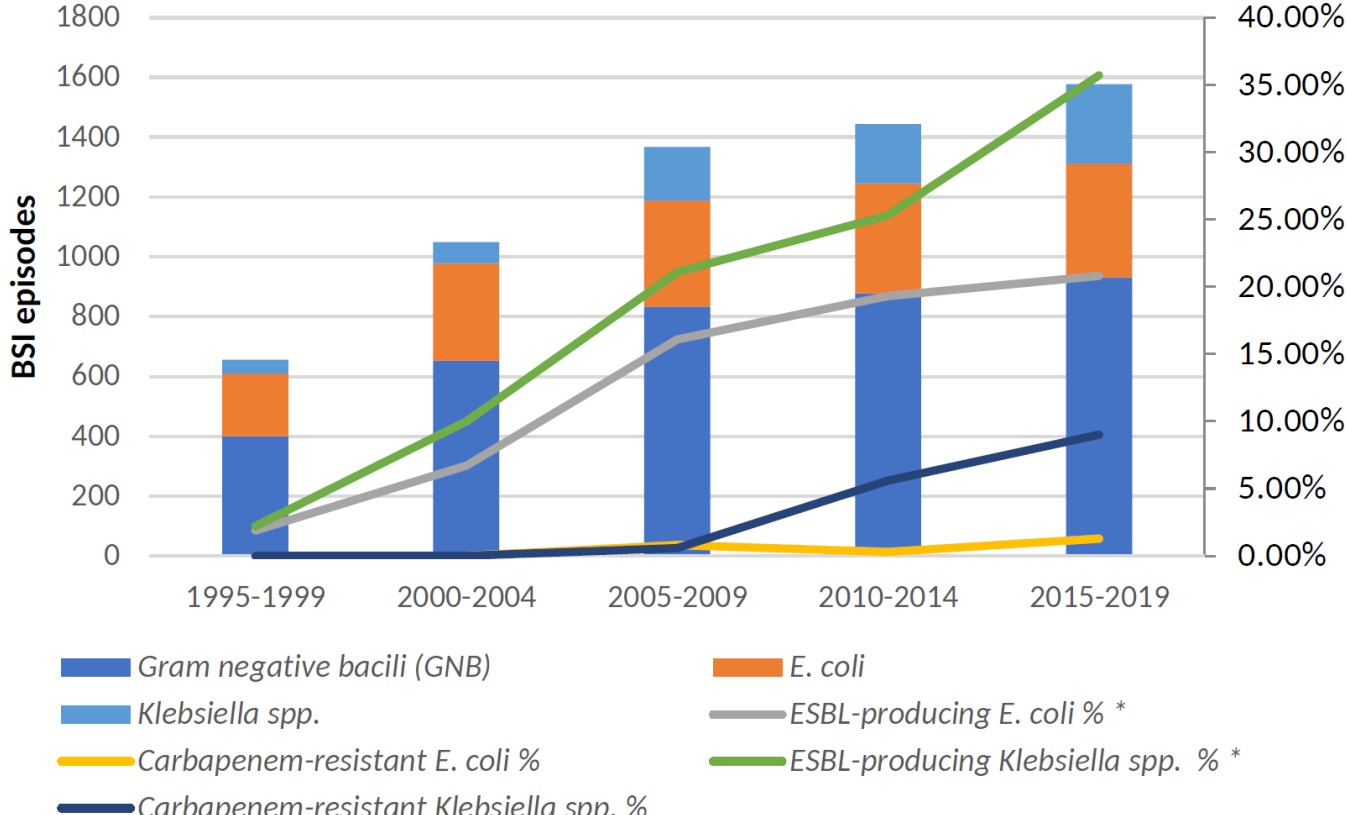

**FIG 2** Changes over time in *E. coli* and *Klebsiella* spp. resistance (including ESBL-producing and carbapenem-resistant isolates). ESBL, extended-spectrum beta-lactamase; MDR, multidrug-resistant; XDR, extensively drug-resistant; and BSI, bloodstream infection.

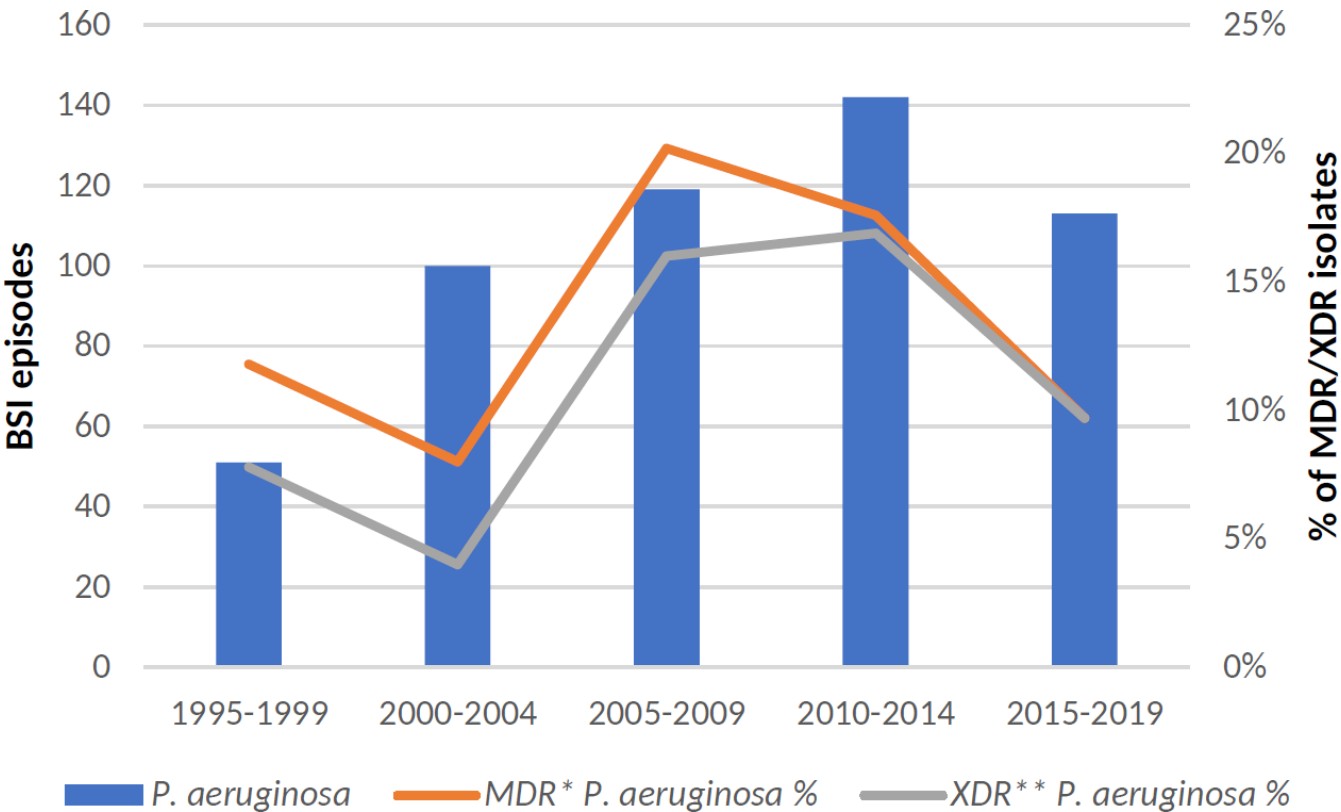

**FIG 3** Changes over time in *P. aeruginosa* resistance to the main antipseudomonal agents. MDR, multidrug-resistant; XDR, extensively drug-resistant; and BSI, bloodstream infection.

## DISCUSSION

This article shows that several changes in BSI epidemiology have occurred in patients with SC over the past decades. These differences represent a current challenge in terms of empiric antibiotic treatment of this population, as made evident by the high percentage of patients with GNB infections who receive IEAT, and elevated 30-day mortality.

The documented changes in BSI epidemiology may be due to various reasons. There are host factors that have changed over the years: in historical studies (2, 6), breast, hepatobiliary, and lung cancer were the most common types of cancer in patients who developed BSI. Here, the most frequent location of SC was colorectal, followed by hepatobiliary and urothelial. Infections often depend on the localization and size of the SC, or the surgeries or medical devices required for its management (18). Important variations in bacteremia sources have also occurred (19). It is worth highlighting the decrease in BSI of pulmonary origin and the increase in infections of urinary origin, in line with the changes in the prevalence of cancer locations. Catheter sepsis is becoming less frequent (20), perhaps due to the better management of indwelling catheters (21). All these changes make it easy to understand the decline in CGP infections, except for *Enterococcus faecium* infection, and the rise of GNB infections, particularly of *Klebsiella* spp.

The increase in *E. faecium* infections has been described previously (22). The possible causes of such emergence have been hypothesized as a result of the increasing use of beta-lactams and, in particular carbapenems due to the high prevalence of ESBL-Enterobacterales. Their low activity on its own against *Enterococci* favors its selection and expansion in the gut, increasing the risk of endogenous bacteremia (23).

**TABLE 3** Treatment and outcomes[e]

| Number of BSI (%) per period | 1995–1999 | 2000–2004 | 2005–2009 | 2010–2014 | 2015–2019 | Overall | P-value |
|---|---|---|---|---|---|---|---|
| | n = 900 (14.7) | n = 1183 (19.3) | n = 1352 (22.1) | n = 1325 (21.7) | n = 1357 (22.2) | n = 6117 (100) | for trend |
| IEAT | 234/900 (26.0) | 288/1183 (24.3) | 358/1352 (26.5) | 355/1325 (26.8) | 285/1319 (21.0) | 1520/6117 (24.8) | NS |
| IEAT Enterobacterales / Enterobacterales BSI | 27/316 (8.5) | 68/512 (13.3) | 119/673 (17.7) | 131/699 (18.7) | 107/779 (13.7) | 452/2979 (15.2) | <0.001 |
| IEAT ESBL / ESBL BSI | 5/15 (33.3) | 15/45 (33.3) | 63/129 (48.8) | 78/145 (53.8) | 66/206 (32.0) | 227/540 (42.0) | <0.001 |
| IEAT carbapenem-resistant/ carbapenem-resistant BSI | 0 (0.0) | 0 (0.0) | 3/4 (75.0) | 5/12 (41.7) | 15/29 (51.7) | 23/45 (51.1) | <0.001 |
| IEAT P. aeruginosa / P. aeruginosa | 9/51 (17.6) | 31/100 (31.0) | 42/119 (35.3) | 49/142 (34.5) | 34/113 (30.1) | 165/525 (31.4) | 0.02 |
| IEAT MDR P. aeruginosa[b,c]/ MDR P. aeruginosa[b,c] | 2/6 (33.3) | 5/8 (62.5) | 14/24 (58.3) | 14/25 (56.0) | 8/11 (72.7) | 43/74 (58.1) | NS |
| IEAT XDR P. aeruginosa[b,c]/ / XDR P. aeruginosa[b,c] | 2/4 (50.0) | 3/4 (75.0) | 12/19 (63.2) | 14/24 (58.3) | 8/11 (72.7) | 39/62 (62.9) | NS |
| **Number of patients with SC[a] (%)[d]** | **1995–1999** | **2000–2004** | **2005–2009** | **2010–2014** | **2015–2019** | **Overall** | **P-value** |
| | 869 (15.0) | 1,139 (19.7) | 1,284 (22.2) | 1,238 (21.4) | 1,257 (21.7) | 5,787 (100) | for trend |
| Related mortality | 133 (15.3) | 126 (11.1) | 128 (10) | 134 (10.8) | 136 (10.8) | 657 (11.4) | 0.009 |
| 30-day mortality | 157 (18.1) | 158 (13.9) | 198 (15.4) | 210 (17.0) | 220 (17.5) | 943 (16.3) | NS |
| 30-day mortality Enterobacterales / 30-day mortality | 34/157 (21.7) | 51/158 (32.3) | 68/198 (34.3) | 68/210 (32.4) | 78/220 (35.5) | 299/943 (31.7) | NS |
| 30-day mortality P. aeruginosa / 30-day mortality | 9/157 (5.7) | 21/158 (13.3) | 17/198 (8.6) | 17/210 (8.1) | 12/220 (5.5) | 76/943 (8.1) | NS |

[a]Abbreviations. MRSA, methicillin-resistant *Staphylococcus aureus*; ESBL, extended-spectrum beta-lactamase; MDR, multidrug-resistant; XDR, extensively drug-resistant; GPC, Gram-positive cocci; GNB, Gram-negative bacilli.
[b]MDR *P. aeruginosa* is defined as non-susceptibility to at least one agent in three or more antimicrobial categories.
[c]XDR *P. aeruginosa*/XDR is defined as non-susceptible to at least one agent in all but two or fewer antimicrobial categories.
[d]Some patients' data were not included due to loss in follow-up (some patients might had been transferred to another hospital or facility).
[e]Antimicrobial categories and agents used to define MDR and XDR *P. aeruginosa* include aminoglycosides, antipseudomonal carbapenems, antipseudomonal cephalosporins, antipseudomonal fluoroquinolones, antipseudomonal penicillins + β-lactamase inhibitors, monobactams, phosphonic acids, and polymyxins.

Empiric coverage of GNB represents a major challenge in treating patients with SC. In our study, *E. coli* was the most frequently isolated pathogen, although there was a sharp increase for *Klebsiella* spp. The main concern associated with these infections has been the growing antimicrobial resistance trend over the years, as reported in other studies (8, 9). Specifically, there was an important rise in ESBL and, more recently, carbapenem-resistant isolates (24, 25). Additionally, the most common antipseudomonal antibiotics were frequently non-active, since more than half of the patients presenting MDR or XDR *P. aeruginosa* infection received inadequate antibiotic treatment (26).

Our study, in line with previous studies (6), documented that those patients receiving IEAT have higher mortality. Accordingly, it should be considered to rethink the empiric antibiotic regimen for these patients. Although there are some treatment guidelines for febrile neutropenia in oncologic patients (27, 28), most patients with cancer and BSI are not neutropenic. The results reported in this research justify the possible creation of a specific guideline to treat such an infection in patients with cancer, even if they do not present neutropenia in the near future. Our article provides data for improving risk assessment for MDR infections, those patients with prior antibiotic therapy, those having BSI while receiving antibiotics, as well as those with BSI from biliary or urinary sources that have a higher risk. Conversely, patients with community-acquired infections exhibit a lower prevalence of MDR BSI.

Specifically, recommendations concerning the best empiric approach for GNB infections should be reassessed, including the role of new antibiotics and/or combination treatments with empiric antibiotics. Considering that the universal use of new antibiotics is not affordable, clinicians should explore innovative ways of identifying patients with a high risk of MDR-GNB BSI or prescribing broad-spectrum antibiotics followed by early de-escalation strategies, with the primary goal of reducing the

inappropriate use of antibiotics in this population, alongside establishing a policy aimed at rationalizing antibiotic usage.

Our study strengths arise from the fact that a senior specialist in infectious diseases collected and evaluated data from a 25-year time span. However, our study has some limitations. It is a single-centre study and data might differ from areas with other microbiologic ecology or varying SC prevalence. The lack of genetic assessment of the strains hindered a more precise description of the clonal changes over time. Our study did not focus on identifying factors associated with mortality, so despite describing an association between IEAT and mortality, other co-factors were not evaluated. Finally, MIC data were not used in our study to define MDROs.

In conclusion, the epidemiology in patients with SC and BSI has changed. An increase in MDR-GNB has compromised the current empiric antibiotic strategies used to date. The percentage of patients currently receiving inadequate empiric treatment is unacceptable, especially given that IEAT is associated with increased mortality. New treatment strategies such as using better tools to stratify patients by risk, prescribing empiric antibiotics with lower resistance rates, and initiating early de-escalation should be promoted.

## ACKNOWLEDGMENTS

We thank OP post-doctoral fellow financial support: La Ligue Nationale contre le Cancer (convention number: AAPMRC 2022/OP) and La Direction de l'Assistance Publique - Hôpitaux de Paris (APHP). P.P.-A. [JR20/00012, PI21/00498, and ICI21/00103] and C.G.-V. [FIS PI21/01640 and ICI21/00103] have received research grants from the Ministerio de Sanidad y Consumo, Instituto de Salud Carlos III and FEDER "Una manera de hacer Europa". Project PI21/01640 has receiving support from Instituto de Salud Carlos III (ISCIII) and is co-funded by the European Union. This work was co-funded by a research grant (SGR 01324 Q5856414G) from the AGAUR (Agencia de Gestión de Ayudas Universitarias y de Investigación) of Catalunya. The funders had no specific role in study design, data collection, writing of the paper, or decision to submit.

We also thank Anthony Armenta for providing medical editing assistance for the manuscript.

## AUTHOR AFFILIATIONS

[1]Infectious Diseases Department, Hospital Clínic de Barcelona, Barcelona, Spain
[2]Emergency Department, Hôpital Saint-Louis, Assistance Publique - Hôpitaux de Paris, Paris, France
[3]Microbiology Department, Hospital Clínic de Barcelona, Barcelona, Spain
[4]Universitat de Barcelona, Barcelona, Spain

## AUTHOR ORCIDs

Tommaso Francesco Aiello http://orcid.org/0000-0003-4441-6318
Mariana Chumbita http://orcid.org/0000-0001-5372-6444
Marta Bodro https://orcid.org/0000-0002-0520-8279
Pedro Puerta-Alcalde http://orcid.org/0000-0003-2490-0217
Carolina Garcia-Vidal http://orcid.org/0000-0002-8915-0683

## FUNDING

| Funder | Grant(s) | Author(s) |
| --- | --- | --- |
| Instituto de Salud Carlos III | JR20/00012, PI21/00498, ICI21/00103 | Pedro Puerta-Alcalde |
| Instituto de Salud Carlos III | PI21/01640, ICI21/00103 | Carolina Garcia-Vidal |
| La Ligue Nationale contre le Cancer | AAPMRC 2022/OP | Olivier Peyrony |

| Funder | Grant(s) | Author(s) |
|---|---|---|
| Agencia de Gestion de Ayudas Universitarias y de Investigacion | SGR 01324 Q5856414G | Carolina Garcia-Vidal |

## AUTHOR CONTRIBUTIONS

Carlos Lopera, Conceptualization, Data curation, Formal analysis, Funding acquisition, Investigation, Methodology, Project administration, Resources, Software, Supervision, Validation, Visualization, Writing – original draft, Writing – review and editing | Patricia Monzó, Conceptualization, Data curation, Investigation, Methodology, Software, Supervision, Validation, Visualization, Writing – original draft, Writing – review and editing | Tommaso Francesco Aiello, Conceptualization, Data curation, Investigation, Methodology, Software, Supervision, Validation, Visualization, Writing – original draft, Writing – review and editing | Mariana Chumbita, Conceptualization, Data curation, Investigation, Methodology, Software, Supervision, Validation, Visualization, Writing – original draft, Writing – review and editing | Olivier Peyrony, Conceptualization, Data curation, Formal analysis, Funding acquisition, Investigation, Methodology, Project administration, Resources, Software, Supervision, Validation, Visualization, Writing – original draft, Writing – review and editing | Antonio Gallardo-Pizarro, Conceptualization, Data curation, Investigation, Methodology, Software, Supervision, Validation, Visualization, Writing – original draft, Writing – review and editing | Cristina Pitart, Conceptualization, Data curation, Investigation, Methodology, Software, Supervision, Validation, Visualization, Writing – original draft, Writing – review and editing | Guillermo Cuervo, Conceptualization, Data curation, Investigation, Methodology, Software, Supervision, Validation, Visualization, Writing – original draft, Writing – review and editing | Laura Morata, Conceptualization, Data curation, Investigation, Methodology, Software, Supervision, Validation, Visualization, Writing – original draft, Writing – review and editing | Marta Bodro, Conceptualization, Data curation, Investigation, Methodology, Software, Supervision, Validation, Visualization, Writing – original draft, Writing – review and editing | Sabina Herrera, Conceptualization, Data curation, Investigation, Methodology, Software, Supervision, Validation, Visualization, Writing – original draft, Writing – review and editing | Ana Del Río, Conceptualization, Data curation, Investigation, Methodology, Software, Supervision, Validation, Visualization, Writing – original draft, Writing – review and editing | José Antonio Martínez, Conceptualization, Data curation, Investigation, Methodology, Software, Supervision, Validation, Visualization, Writing – original draft, Writing – review and editing | Alex Soriano, Conceptualization, Data curation, Investigation, Methodology, Software, Supervision, Validation, Visualization, Writing – original draft, Writing – review and editing | Pedro Puerta-Alcalde, Conceptualization, Data curation, Formal analysis, Funding acquisition, Investigation, Methodology, Project administration, Resources, Software, Supervision, Validation, Visualization, Writing – original draft, Writing – review and editing | Carolina Garcia-Vidal, Conceptualization, Data curation, Formal analysis, Funding acquisition, Investigation, Methodology, Project administration, Resources, Software, Supervision, Validation, Visualization, Writing – original draft, Writing – review and editing

## ADDITIONAL FILES

The following material is available online.

Open Peer Review

**PEER REVIEW HISTORY (review-history.pdf).** An accounting of the reviewer comments and feedback.

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
