## [Reviewer comments · Microbiology Spectrum]

Microbiology Spectrum

Prevalence and Impact of Multidrug-Resistant Bacteria in Solid Cancer Patients with Bloodstream Infection: A 25-Year Trend Analysis.

Carlos Lopera, Patricia Monzó, Tommaso Francesco Aiello, Mariana Chumbita, Oliver Peyrony, Antonio Gallardo, Cristina Pitart, Guillermo Cuervo, Laura Morata, Marta Bodro, sabina herrera, Ana del Río, José Martínez, Alex Soriano, Pedro Puerta-Alcalde, and Carolina García-Vidal

Corresponding Author(s): Carolina García-Vidal, Department of Infectious Diseases, Hospital Clínic de Barcelona

Review Timeline:

Submission Date:	October 19, 2023
Editorial Decision:	November 29, 2023
Revision Received:	January 21, 2024
Editorial Decision:	February 6, 2024
Revision Received:	February 27, 2024
Accepted:	March 14, 2024

Editor: Ahmed Babiker

Reviewer(s): The reviewers have opted to remain anonymous.

Transaction Report:

DOI: <https://doi.org/10.1128/spectrum.02961-23>

Re: Spectrum02961-23 (Prevalence and Impact of Multidrug-Resistant Bacteria in Solid Cancer Patients with Bloodstream Infection: A 25-Year Trend Analysis.)

Dear Dr. Carolina García-Vidal:

Thank you for the privilege of reviewing your work. Below you will find my comments, instructions from the Spectrum editorial office, and the reviewer comments.

Editor Comments:

The authors mention that Chi square for trends was used to compare the different time spans. It seems to me that a more appropriate way to assess the statistical significance of The MDRO trends over time would be better analyzed by converting time(year) into a continuous variable and either performing a linear regression or Poisson/binomial over time with MDRO trends expressed as a rates

Other limitations should be mentioned in the limitations paragraph. Breakpoints have changed over time (especially for GNRs) and authors have not used MIC data and standardized definition to defined MDROs. No multivariable analysis was performed when looking at the relationship of IEAT and mortality. While its is likely a true signal, other confounders should have been accounted for.

Revision Guidelines

Sincerely,
Ahmed Babiker
Editor
Microbiology Spectrum

Reviewer #1 (Comments for the Author):

Thank you very much for allowing me to review the manuscript "Prevalence and Impact of Multidrug-Resistant Bacteria in Solid Cancer Patients with Bloodstream Infection: A 25-Year Trend Analysis." This article is mainly a descriptive study retrospectively reviewing the changing trends in multidrug resistant bacterial bloodstream infections at their medical center, and the rates of inappropriate empiric antibiotic selection. The authors demonstrate that rates of MDR Gram negative bacteremia are on the rise, and the rates of inappropriate empiric antibiotic use are also on the rise. The authors rightly call on the community to investigate a set of risk criteria for which we can use to understand patients at risk for MDR BSI and perhaps qualify for novel BL/BLI regimens upfront.

Major Comments:

My major comment is for the authors to please consider looking at *who* developed MDR bloodstream infections. Was it patients who had previous episodes of BSI? Was it patients of a certain age group? Patients with hospital acquired infections? Patients who had received antibiotic therapy in the last 6 months? Patients on antibiotic prophylaxis? I think this is important to capture who we can identify in the future as being at risk for MDR BSI and ensure they are receiving appropriate antibiotics.

Minor comments:

Abstract: Would add a little more information to the methods as able, within the constraints of the word count limitations, to include which MDR infections were studied and how the authors defined the different time periods, etc.

Methods: Please define neutropenia (what cutoff did the authors use)

Methods: For the Coagulase negative staph bloodstream infections, how did the authors determine if these were true bloodstream infection vs contaminant? Would include this criteria in the methods.

Table 1: Prior corticosteroid use: What was the time range considered for this? Would include as a footnote 'corticosteroid use within the preceding XX months prior to BSI'

Discussion: One of the reasons for MDR infections is the use of inappropriately broad antibiotics. Would include some sort of statement that addresses the importance of balancing stewardship of broad-spectrum antibiotics to try to address the rise of MDROs.

Reviewer #2 (Comments for the Author):

Dear Editor of Microbiology Spectrum,

Thank you for the opportunity to review the manuscript "Prevalence and Impact of Multidrug-Resistant Bacteria in Solid Cancer Patients with Bloodstream Infection: A 25-Year Trend Analysis". In the manuscript, the authors examine the epidemiology of multidrug-resistant bacteria in solid cancer patients with bloodstream infection over a 25-year period at a university hospital. The study offers valuable insights into the changing ecology of MDR bacteria in the hospital setting, highlighting the increase in Gram-negative bacilli, especially ESBL-producing strains, and the associated mortality risks with inappropriate empiric antibiotic treatment.

Overall, the study is well-executed and written, and presents findings that contribute to our understanding of the infections that affect cancer patients. Despite being a single-centre study, the cohort followed is quite large and is followed over an extended period. I also wish to congratulate the authors on a clear, descriptive design without unnecessarily complex statistical analyses, which effectively illuminates the evolving patterns of antimicrobial resistance.

Detailed comments and suggestions for further improvement of the manuscript are provided in the following sections.

Major comments

Overall

The authors have included a large number of p-values, seemingly for descriptive purposes to indicate the precision of measurements in the study, rather than to draw inferences. However, this inadvertently leads to a multiple comparisons problem, with a high risk of false positive findings. A more descriptive measure of precision would be more suitable for this purpose. It would be beneficial to limit hypothesis testing to critical comparisons related to the main argument of the study (the aim and conclusion), in particular the relationship between inappropriate empiric antibiotic treatment and increased mortality rates, which is also not found in the tables.

Discussion, line 197-199

The manuscript states that the shift in BSIs from Gram-positive to Gram-negative bacteria might be influenced by the underlying cancer disease. However, this raises a question of the causal direction suggested. Given the known rise of resistant Gram-negative bacteria in healthcare settings across Europe and the successful prevention of resistant Gram-positive bacteria in many countries, it is important to consider whether the shift in epidemiology precedes and influences the types of BSIs observed in cancer patients. If the broader epidemiological trend is indeed the driving factor, one would expect to see an increase in BSIs caused by resistant Gram-negative bacteria in cancers that can cause an exposure to the ecological niches of these bacteria, such as the gut. Conversely, there should be a decrease in BSIs caused by Gram-positive bacteria in cancers typically treated surgically, where post-surgical infections are more likely. If the authors agree that this reverse direction is possible, it should be mentioned.

Discussion, line 214-220

The study suggests the need for specific guidelines for patients with cancer based on its findings. However, this argument lacks a comparison between common solid cancer patient groups, such as those with breast or colorectal cancer, and the average BSI patient without malignancy. Without this comparison, the results could potentially be interpreted as reflective of the deteriorating epidemiology in the general patient population, rather than indicative of a unique trend within the cancer patient subgroup.

Discussion

A central aspect of the study involves the comparison between inappropriate and appropriate empiric antibiotic treatments, with some patients with rather extensively resistant bugs end up with an appropriate empiric treatment. To enhance the understanding, especially for readers from different countries, it would be beneficial to include some examples and a description of the antibiotic selection process in your hospital.

Minor

Abstract, results

Consider adding the denominator for all percentages so that readers can more easily follow how the data is grouped.

Importance

Should be revised again after all other changes are done.

Introduction, line 68-69

Consider using more measured terms such instead of "spectacular" and "devastating" to describe survival improvements, for a tone more fitting to a scientific article. Also, some cancers have had a more steady and/or incremental improvement in survival.

Introduction, line 75

Can the reason for why a "closer relationship with" (or exposure to) the healthcare environment is relevant be detailed briefly?

Introduction, line 77

The aim of "describing the epidemiology" is very broad, perhaps specify that it primarily refers to the microbial distribution, as incidence and clinical features are distinctly mentioned elsewhere.

Introduction, line 79

Detailed the impact of the changes, or the impact of IEAT?

Introduction

As a suggestion, you might find Rolston KV. Infections in Cancer Patients with Solid Tumors: A Review. *Infect Dis Ther.* 2017 Mar;6(1):69-83. doi: 10.1007/s40121-017-0146-1. Epub 2017 Feb 3. PMID: 28160269; PMCID: PMC5336421 quite informative for describing mechanisms of infectious complications in solid cancer patients, and it may add diversity to the current set of references which seems to primarily come from the same research community.

Methods, line 98

How many patients switched antibiotics within those first 24 hours? This timeframe may be quite broad.

Methods, line 101

Where is the mortality data from? Are you able to follow patients for mortality outside of the hospital?

Methods, line 93-94

Consider explaining the rationale behind dividing the follow-up period into five-year intervals, as treating the time period as continuous might have yielded a stronger analysis.

Related to Table 1

There is quite a large proportion of patients with an unknown focus of bloodstream infection - what bias is this likely to cause? E.g., is a urinary focus more often correctly classified than a catheter-related bloodstream infection?

Results

Enhance the readability of central numbers in the tables by providing both the numerator and denominator for each percentage, allowing readers to more easily follow data groupings.

Results, line 157

Were not any MDRs part of a polymicrobial BSI?

Table 2

Several points that could benefit from clarification to aid reader comprehension:

1. There may be something I am misunderstanding in this table, as the listing of Gram-positive cocci species summed total (CoNS 701, Enterococci 584, *S. aureus* 470, Streptococci 464) equals 2219, which is 37.7% of the total 5896 BSI episodes, not the 2165 (36.7%) listed as Gram-positive cocci. Please clarify if the numbers in the table is correct and if so, how they should be interpreted.
2. While it's helpful to have subcategories with resistance proportions, the format should clearly indicate that percentages for resistant strains (e.g., MRSA) are calculated using their broader category (e.g., *S. aureus*) as the denominator. This also goes for Gram-positive species which are included in the Gram-positive cocci supercategory.
3. The total percentages for Gram-positive cocci, Gram-negative bacilli, and *Candida* spp. amount to 101.4%, which raises questions. This exceeds the 100% expected if each BSI is represented by one microbe, yet it's not sufficient to include all polymicrobial BSIs, which account for 10.6%. Are there microbial categories not represented in the table, perhaps warranting an "other" category, or am I misunderstanding something?
4. The microbial distribution within polymicrobial BSIs would be valuable to understand the complete picture of the infections studied.
5. Finally, could you elaborate on the criteria used to distinguish when CoNS are considered contaminants versus pathogens? Are CoNS included in the polymicrobial BSI counts, and if so, how is this determination made?

Figure 1 and 2

1. Figure 1 could be considered converted into a stacked bar chart. This format would more clearly illustrate that *E. coli* and *Klebsiella* spp. are part of the overall total of microorganisms.
2. The time line in the graph should extend from 1995 to 2019, rather than from 1995 to 'overall.' Using a line graph to represent continuous time makes the inclusion of an 'overall' bar graphically inconsistent. Ending the line in 2019 would more effectively demonstrate the increasing trend in resistance, culminating in its highest level at the end of the observed period.
3. Ensure that all abbreviations used in the figures are explained in their respective captions for clarity. This is particularly important for Figure 2, where readers need to understand the main antipseudomonal agents and the definitions of resistance patterns, such as MDR and XDR.

Table 3

Include numbers in the column headers for clarity. Additionally, ensure that the denominators for the percentages are clearly indicated, as it's currently not always apparent what they are based on.

Discussion, line 184

Do you mean statistically significant or clinically significant? Consider using a more specific term to avoid ambiguity.

Discussion, line 180-191

An important backdrop is whether your cancer population is representative of the broader cancer population in the city, region, or country? Or is the cancer population at this particular hospital a selected population with e.g., fewer breast cancers?

Discussion, line 200

A reference is needed to support the statement made here.

Discussion, line 203

Reconsider the use of the word "intrinsic"; perhaps "on its own" or a similar phrase would be more appropriate.

Discussion, line 209

Clarify whether "series" is intended to mean "studies."

Discussion, line 229-230

This appears to be an incomplete sentence.

Dear Editor of Microbiology Spectrum,

Thank you for the opportunity to review the manuscript "Prevalence and Impact of Multidrug-Resistant Bacteria in Solid Cancer Patients with Bloodstream Infection: A 25-Year Trend Analysis". In the manuscript, the authors examine the epidemiology of multidrug-resistant bacteria in solid cancer patients with bloodstream infection over a 25-year period at a university hospital. The study offers valuable insights into the changing ecology of MDR bacteria in the hospital setting, highlighting the increase in Gram-negative bacilli, especially ESBL-producing strains, and the associated mortality risks with inappropriate empiric antibiotic treatment.

Overall, the study is well-executed and written, and presents findings that contribute to our understanding of the infections that affect cancer patients. Despite being a single-centre study, the cohort followed is quite large and is followed over an extended period. I also wish to congratulate the authors on a clear, descriptive design without unnecessarily complex statistical analyses, which effectively illuminates the evolving patterns of antimicrobial resistance.

Detailed comments and suggestions for further improvement of the manuscript are provided in the following sections.

Major comments

Overall

The authors have included a large number of p-values, seemingly for descriptive purposes to indicate the precision of measurements in the study, rather than to draw inferences. However, this inadvertently leads to a multiple comparisons problem, with a high risk of false positive findings. A more descriptive measure of precision would be more suitable for this purpose. It would be beneficial to limit hypothesis testing to critical comparisons related to the main argument of the study (the aim and conclusion), in particular the relationship between inappropriate empiric antibiotic treatment and increased mortality rates, which is also not found in the tables.

Discussion, line 197-199

The manuscript states that the shift in BSIs from Gram-positive to Gram-negative bacteria might be influenced by the underlying cancer disease. However, this raises a question of the causal direction suggested. Given the known rise of resistant Gram-negative bacteria in healthcare settings across Europe and the successful prevention of resistant Gram-positive bacteria in many countries, it is important to consider whether the shift in epidemiology precedes and influences the types of BSIs observed in cancer patients. If the broader epidemiological trend is indeed the driving factor, one would expect to see an increase in BSIs caused by resistant Gram-negative bacteria in cancers that can cause an exposure to the ecological niches of these bacteria, such as the gut. Conversely, there should be a decrease in BSIs caused by Gram-positive bacteria in cancers typically treated surgically, where post-surgical infections are more likely. If the authors agree that this reverse direction is possible, it should be mentioned.

Discussion, line 214-220

The study suggests the need for specific guidelines for patients with cancer based on its findings. However, this argument lacks a comparison between common solid cancer patient groups, such as those with breast or colorectal cancer, and the average BSI patient without malignancy. Without this comparison, the results could potentially be interpreted as reflective of the deteriorating epidemiology in the general patient population, rather than indicative of a unique trend within the cancer patient subgroup.

Discussion

A central aspect of the study involves the comparison between inappropriate and appropriate empiric antibiotic treatments, with some patients with rather extensively resistant bugs end up with an appropriate empiric treatment. To enhance the understanding, especially for readers from different countries, it would be beneficial to include some examples and a description of the antibiotic selection process in your hospital.

Minor

Abstract, results

Consider adding the denominator for all percentages so that readers can more easily follow how the data is grouped.

Importance

Should be revised again after all other changes are done.

Introduction, line 68-69

Consider using more measured terms such instead of "spectacular" and "devastating" to describe survival improvements, for a tone more fitting to a scientific article. Also, some cancers have had a more steady and/or incremental improvement in survival.

Introduction, line 75

Can the reason for why a "closer relationship with" (or exposure to) the healthcare environment is relevant be detailed briefly?

Introduction, line 77

The aim of "describing the epidemiology" is very broad, perhaps specify that it primarily refers to the microbial distribution, as incidence and clinical features are distinctly mentioned elsewhere.

Introduction, line 79

Detailed the impact of the changes, or the impact of IEAT?

Introduction

As a suggestion, you might find *Rolston KV. Infections in Cancer Patients with Solid Tumors: A Review. Infect Dis Ther. 2017 Mar;6(1):69-83. doi: 10.1007/s40121-017-0146-1. Epub 2017 Feb 3. PMID: 28160269; PMCID: PMC5336421* quite informative for describing mechanisms of infectious complications in solid cancer patients, and it may add diversity to the current set of references which seems to primarily come from the same research community.

Methods, line 98

How many patients switched antibiotics within those first 24 hours? This timeframe may be quite broad.

Methods, line 101

Where is the mortality data from? Are you able to follow patients for mortality outside of the hospital?

Methods, line 93-94

Consider explaining the rationale behind dividing the follow-up period into five-year intervals, as treating the time period as continuous might have yielded a stronger analysis.

Related to Table 1

There is quite a large proportion of patients with an unknown focus of bloodstream infection – what bias is this likely to cause? E.g., is a urinary focus more often correctly classified than a catheter-related bloodstream infection?

Results

Enhance the readability of central numbers in the tables by providing both the numerator and denominator for each percentage, allowing readers to more easily follow data groupings.

Results, line 157

Were not any MDRs part of a polymicrobial BSI?

Table 2

Several points that could benefit from clarification to aid reader comprehension:

1. There may be something I am misunderstanding in this table, as the listing of Gram-positive cocci species summed total (CoNS 701, Enterococci 584, *S. aureus* 470, Streptococci 464) equals 2219, which is 37.7% of the total 5896 BSI episodes, not the 2165 (36.7%) listed as Gram-positive cocci. Please clarify if the numbers in the table is correct and if so, how they should be interpreted.
2. While it's helpful to have subcategories with resistance proportions, the format should clearly indicate that percentages for resistant strains (e.g., MRSA) are calculated using their broader category (e.g., *S. aureus*) as the denominator. This also goes for Gram-positive species which are included in the Gram-positive cocci supercategory.
3. The total percentages for Gram-positive cocci, Gram-negative bacilli, and *Candida* spp. amount to 101.4%, which raises questions. This exceeds the 100% expected if each BSI is represented by one microbe, yet it's not sufficient to include all polymicrobial BSIs, which account for 10.6%. Are there microbial categories not represented in the table, perhaps warranting an "other" category, or am I misunderstanding something?
4. The microbial distribution within polymicrobial BSIs would be valuable to understand the complete picture of the infections studied.
5. Finally, could you elaborate on the criteria used to distinguish when CoNS are considered contaminants versus pathogens? Are CoNS included in the polymicrobial BSI counts, and if so, how is this determination made?

Figure 1 and 2

1. Figure 1 could be considered converted into a stacked bar chart. This format would more clearly illustrate that *E. coli* and *Klebsiella* spp. are part of the overall total of microorganisms.
2. The time line in the graph should extend from 1995 to 2019, rather than from 1995 to 'overall.' Using a line graph to represent continuous time makes the inclusion of an 'overall' bar graphically inconsistent. Ending the line in 2019 would more effectively demonstrate the increasing trend in resistance, culminating in its highest level at the end of the observed period.
3. Ensure that all abbreviations used in the figures are explained in their respective captions for clarity. This is particularly important for Figure 2, where readers need to understand the main antipseudomonal agents and the definitions of resistance patterns, such as MDR and XDR.

Table 3

Include numbers in the column headers for clarity. Additionally, ensure that the denominators for the percentages are clearly indicated, as it's currently not always apparent what they are based on.

Discussion, line 184

Do you mean statistically significant or clinically significant? Consider using a more specific term to avoid ambiguity.

Discussion, line 180-191

An important backdrop is whether your cancer population is representative of the broader cancer population in the city, region, or country? Or is the cancer population at this particular hospital a selected population with e.g., fewer breast cancers?

Discussion, line 200

A reference is needed to support the statement made here.

Discussion, line 203

Reconsider the use of the word "intrinsic"; perhaps "on its own" or a similar phrase would be more appropriate.

Discussion, line 209

Clarify whether "series" is intended to mean "studies."

Discussion, line 229-230

This appears to be an incomplete sentence.

Editor and Reviewer's comments:

Editor Comments:

The authors mention that Chi square for trends was used to compare the different time spans. It seems to me that a more appropriate way to assess the statistical significance of The MDRO trends over time would be better analyzed by converting time(year) into a continuous variable and either performing a linear regression or Poisson/binomial over time with MDRO trends expressed as a rates.

R: First, we would like to express our gratitude to the editor and reviewers for their excellent work, which we believe has significantly enhanced our article. We acknowledge the editor's comment in line with the reviewers' suggestions for a more comprehensive description of multidrug resistance risk factors. We hope that in revising the article, we have adequately addressed this requirement. With this aim, we have integrated a logistic regression analysis to identify MDR risk factors, incorporating the temporal period as a co-factor. It is our perception that existing literature has primarily focused on delineating factors associated with MDR in terms of the dichotomous variable of MDR or non-MDR, rather than the temporal percentage, which serves as the dependent variable in linear regression or Poisson. We present this revised version accordingly but remain open to further considerations should the editor deem it necessary. (Please see next comments and new article version).

Other limitations should be mentioned in the limitations paragraph. Breakpoints have changed over time (especially for GNRs) and authors have not used MIC data and standardized definition to defined MDROs. No multivariable analysis was performed when looking at the relationship of IEAT and mortality. While its is likely a true signal, other confounders should have been accounted for.

R: Following the editor's recommendation, we have included these limitations in the article (Please see discussion, lines 305-311, page 14).

Reviewer #1 (Comments for the Author):

Thank you very much for allowing me to review the manuscript "Prevalence and Impact of Multidrug-Resistant Bacteria in Solid Cancer Patients with Bloodstream Infection: A 25-Year Trend Analysis." This article is mainly a descriptive study retrospectively reviewing the changing trends in multidrug resistant bacterial bloodstream infections at their medical center, and the rates of inappropriate empiric antibiotic selection. The authors demonstrate that rates of MDR Gram negative bacteremia are on the rise, and the rates of inappropriate empiric antibiotic use are also on the rise. The authors rightly call on the community to investigate a set of risk

criteria for which we can use to understand patients at risk for MDR BSI and perhaps qualify for novel BL/BLI regimens upfront.

Major Comments:

My major comment is for the authors to please consider looking at *who* developed MDR bloodstream infections. Was it patients who had previous episodes of BSI? Was it patients of a certain age group? Patients with hospital acquired infections? Patients who had received antibiotic therapy in the last 6 months? Patients on antibiotic prophylaxis? I think this is important to capture who we can identify in the future as being at risk for MDR BSI and ensure they are receiving appropriate antibiotics.

R: We appreciate the reviewer's valuable input that enriches our study. Thank you. Following your guidance, we have conducted a logistic regression analysis to identify the independent risk factors associated with developing MDR bloodstream infection in our cohort. We have integrated this information accordingly in the paper (Please see new abstract, the Methods section, page 9, lines 184-190; Results section, pages 11-12, lines 238-247; Discussion section, page 14, lines 289-293.)

Minor comments:

Abstract: Would add a little more information to the methods as able, within the constraints of the word count limitations, to include which MDR infections were studied and how the authors defined the different time periods, etc.

R: We appreciate your comment and we have included the required information. (Please see abstract, methods).

Methods: Please define neutropenia (what cutoff did the authors use)

R: Done (Please see line 141, page 7).

Methods: For the Coagulase negative staph bloodstream infections, how did the authors determine if these were true bloodstream infection vs contaminant? Would include this criteria in the methods.

R: Done (Please see lines 160-164, page 8).

Table 1: Prior corticosteroid use: What was the time range considered for this? Would include as a footnote 'corticosteroid use within the preceding XX months prior to BSI'.

R: The use of corticosteroids was assessed in the preceding 3 months. This information has been added (Please see new table 1).

Discussion: One of the reasons for MDR infections is the use of inappropriately broad antibiotics. Would include some sort of statement that addresses the importance of

balancing stewardship of broad-spectrum antibiotics to try to address the rise of MDROs.

R: Thank you very much for this value comment. We have added a statement advocating for that balance. (Please see lines 299-301, page 14, discussion section).

Reviewer #2 (Comments for the Author):

Dear Editor of Microbiology Spectrum,

Thank you for the opportunity to review the manuscript "Prevalence and Impact of Multidrug-Resistant Bacteria in Solid Cancer Patients with Bloodstream Infection: A 25-Year Trend Analysis". In the manuscript, the authors examine the epidemiology of multidrug-resistant bacteria in solid cancer patients with bloodstream infection over a 25-year period at a university hospital. The study offers valuable insights into the changing ecology of MDR bacteria in the hospital setting, highlighting the increase in Gram-negative bacilli, especially ESBL-producing strains, and the associated mortality risks with inappropriate empiric antibiotic treatment.

Overall, the study is well-executed and written, and presents findings that contribute to our understanding of the infections that affect cancer patients. Despite being a single-centre study, the cohort followed is quite large and is followed over an extended period. I also wish to congratulate the authors on a clear, descriptive design without unnecessarily complex statistical analyses, which effectively illuminates the evolving patterns of antimicrobial resistance.

Detailed comments and suggestions for further improvement of the manuscript are provided in the following sections.

Major comments

Overall

The authors have included a large number of p-values, seemingly for descriptive purposes to indicate the precision of measurements in the study, rather than to draw inferences. However, this inadvertently leads to a multiple comparisons problem, with a high risk of false positive findings. A more descriptive measure of precision would be more suitable for this purpose. It would be beneficial to limit hypothesis testing to critical comparisons related to the main argument of the study (the aim and conclusion), in particular the relationship between inappropriate empiric antibiotic treatment and increased mortality rates, which is also not found in the tables.

R: With utmost respect to the reviewer's comment, with which we wholeheartedly agree, the quantity of p-values is extremely high, the objective of our study was to delineate the temporal changes in the epidemiology of multidrug-resistant (MDR) bacteria (Table 2), the percentages of inappropriate empiric antibiotic treatment (Table 3) use, and mortality over time (Table 3). In response to the reviewer's comment, we attempted to simplify the results by substituting "p" with "NS" for ease of comprehension. Should the reviewer require any further modifications, we are more than willing to accommodate their requests.

Discussion, line 197-199

The manuscript states that the shift in BSIs from Gram-positive to Gram-negative bacteria might be influenced by the underlying cancer disease. However, this raises a question of the causal direction suggested. Given the known rise of resistant Gram-negative bacteria in healthcare settings across Europe and the successful prevention of resistant Gram-positive bacteria in many countries, it is important to consider whether the shift in epidemiology precedes and influences the types of BSIs observed in cancer patients. If the broader epidemiological trend is indeed the driving factor, one would expect to see an increase in BSIs caused by resistant Gram-negative bacteria in cancers that can cause an exposure to the ecological niches of these bacteria, such as the gut. Conversely, there should be a decrease in BSIs caused by Gram-positive bacteria in cancers typically treated surgically, where post-surgical infections are more likely. If the authors agree that this reverse direction is possible, it should be mentioned.

R: Thank you very much for this insightful comment. We have added a sentence to the discussion that encapsulates this concept, also well reflected by the article recommended by the reviewer (Rolston KV et al.). (Please see line 259-260, page 12; Discussion section and reference 18).

Discussion, line 214-220

The study suggests the need for specific guidelines for patients with cancer based on its findings. However, this argument lacks a comparison between common solid cancer patient groups, such as those with breast or colorectal cancer, and the average BSI patient without malignancy. Without this comparison, the results could potentially be interpreted as reflective of the deteriorating epidemiology in the general patient population, rather than indicative of a unique trend within the cancer patient subgroup.

R: We believe that the revisions made in our article contribute to a better delineation in the current version of patients who would benefit from enhanced antibiotic guidelines (Please see discussion section, page 14, lines 289-293).

Discussion

A central aspect of the study involves the comparison between inappropriate and

appropriate empiric antibiotic treatments, with some patients with rather extensively resistant bugs end up with an appropriate empiric treatment. To enhance the understanding, especially for readers from different countries, it would be beneficial to include some examples and a description of the antibiotic selection process in your hospital.

R: Done (Please see lines 135-140, page 7; methods section).

Minor

Abstract, results: Consider adding the denominator for all percentages so that readers can more easily follow how the data is grouped.

R: Done (Please see abstract; results section).

Importance

Should be revised again after all other changes are done.

R: Done (Please see Importance section).

Introduction, line 68-69

Consider using more measured terms such instead of "spectacular" and "devastating" to describe survival improvements, for a tone more fitting to a scientific article. Also, some cancers have had a more steady and/or incremental improvement in survival.

R: Done (Please see lines 99&101, page 5; Introduction section).

Introduction, line 75

Can the reason for why a "closer relationship with" (or exposure to) the healthcare environment is relevant be detailed briefly?

R: Done (Please see line 107-110, pages 5&6; Introduction section).

Introduction, line 77

The aim of "describing the epidemiology" is very broad, perhaps specify that it primarily refers to the microbial distribution, as incidence and clinical features are distinctly mentioned elsewhere.

R: Done (Please see lines 112-113, page 6; Introduction section).

Introduction, line 79

Detailed the impact of the changes, or the impact of IEAT?

R: To avoid confusion, we rewrote the sentence. (Please see lines 114-115, page 6; Introduction section).

Introduction

As a suggestion, you might find Rolston KV. Infections in Cancer Patients with Solid Tumors: A Review. Infect Dis Ther. 2017 Mar;6(1):69-83. doi: 10.1007/s40121-017-0146-1. Epub 2017 Feb 3. PMID: 28160269; PMCID: PMC5336421 quite informative for describing mechanisms of infectious complications in solid cancer patients, and it may add diversity to the current set of references which seems to primarily come from the same research community.

R: Thank you very much for the recommendation; it was truly helpful. We have incorporated some concepts into the discussion and included this new reference. (Please see lines 259-260, page 12; Discussion section and Reference 18).

Methods, line 98

How many patients switched antibiotics within those first 24 hours? This timeframe may be quite broad.

R: We agree with the reviewer's comment. To avoid confusion, we rephrased the sentence. (Please see line 143, page 7; Methods section).

Methods, line 101

Where is the mortality data from? Are you able to follow patients for mortality outside of the hospital?

R: Mortality was defined as death due to any cause <30 days after BSI. We include this definition in method section (Please see line 155-156, page 8; Methods section).

Methods, line 93-94

Consider explaining the rationale behind dividing the follow-up period into five-year intervals, as treating the time period as continuous might have yielded a stronger analysis.

R: In previous articles published by our group analyzing changes over time (1-3), we always have employed five-year time spans as we believe it facilitates data visualization for the reader. References:

1. Simonetti AF, Garcia-Vidal C, Viasus D, García-Somoza D, Dorca J, Gudiol F, Carratalà J. Declining mortality among hospitalized patients with community-acquired pneumonia. *Clin Microbiol Infect.* 2016 Jun;22(6):567.e1-7. doi: 10.1016/j.cmi.2016.03.015. Epub 2016 Mar 26. PMID: 27021421.
2. Puerta-Alcalde P, Cardozo C, Marco F, Suárez-Lledó M, Moreno E, Morata L, Fernández-Avilés F, Gutiérrez-García G, Chumbita M, Rosiñol L, Martínez JA, Martínez C, Mensa J, Urbano Á, Rovira M, Soriano A, Garcia-Vidal C. Changing epidemiology of bloodstream infection in a 25-years hematopoietic stem cell transplant program: current challenges and pitfalls on empiric antibiotic treatment impacting outcomes. *Bone Marrow Transplant.* 2020 Mar;55(3):603-612. doi: 10.1038/s41409-019-0701-3. Epub 2019 Sep 30. PMID: 31570779.
3. Cillóniz C, Liapikou A, Martin-Loeches I, García-Vidal C, Gabarrús A, Ceccato A, Magdaleno D, Mensa J, Marco F, Torres A. Twenty-year trend in mortality among hospitalized patients with pneumococcal community-acquired pneumonia. *PLoS One.* 2018 Jul 18;13(7):e0200504. doi: 10.1371/journal.pone.0200504. PMID: 30020995; PMCID: PMC6051626.

Related to Table 1

There is quite a large proportion of patients with an unknown focus of bloodstream infection - what bias is this likely to cause? E.g., is a urinary focus more often correctly classified than a catheter-related bloodstream infection?

R: Thank you for your pertinent comment. We acknowledge that the figures for the "unknown" category might appear high, but it's important to note that this category includes patients with endogenous bacteremia, a clarification we've made in the table (Please, see new table 1).

Regarding the analysis of the infection focus, it's worth mentioning that our assessment was conducted prospectively by a specialized medical team with extensive clinical experience in attending to these patients. This team has a strong background in infectious diseases, with numerous publications, which adds credibility to our findings.

Results

Enhance the readability of central numbers in the tables by providing both the numerator and denominator for each percentage, allowing readers to more easily follow data groupings.

R: Done, see Tables 1,2 and 3.

Results, line 157

Were not any MDRs part of a polymicrobial BSI?

R: There were 114 cases of polymicrobial infections that included at least one MDR pathogen.

Table 2

Several points that could benefit from clarification to aid reader comprehension:

1. There may be something I am misunderstanding in this table, as the listing of Gram-positive cocci species summed total (CoNS 701, Enterococci 584, *S. aureus* 470, Streptococci 464) equals 2219, which is 37.7% of the total 5896 BSI episodes, not the 2165 (36.7%) listed as Gram-positive cocci. Please clarify if the numbers in the table is correct and if so, how they should be interpreted.

R: We sincerely apologize for the misunderstanding. We clarify this point. In this latest version, the accurate data is well described. Table 2 has been updated. Polymicrobial infections may be composed of GNB, GPC, and Candida in the detailed proportions (Overall there were 617 polymicrobial BSI, classified as follows: 263 were GPC and GNB infections, 69 were caused by two or more GPC, 4 were polymicrobial GPC (including at least two GPC) and at least one GNB at the same time, 12 episodes were caused by candidemia and GPC, 15 episodes were caused by candidemia and GNB, and 4 episodes of candidemia included at least two different species of Candida, there were also 250 episodes of polymicrobial GNB infections), with the current sum being accurate and revised. See Results section page 10, lines 204-210 and new Table 2.

2. While it's helpful to have subcategories with resistance proportions, the format should clearly indicate that percentages for resistant strains (e.g., MRSA) are calculated using their broader category (e.g., *S. aureus*) as the denominator. This also goes for Gram-positive species which are included in the Gram-positive cocci supercategory.

R: Done, see Table 2.

3. The total percentages for Gram-positive cocci, Gram-negative bacilli, and *Candida* spp. amount to 101.4%, which raises questions. This exceeds the 100% expected if each BSI is represented by one microbe, yet it's not sufficient to include all polymicrobial BSIs, which account for 10.6%. Are there microbial categories not represented in the table, perhaps warranting an "other" category, or am I misunderstanding something?

R: All data has been reviewed and clarified to make it more understandable.

4. The microbial distribution within polymicrobial BSIs would be valuable to understand the complete picture of the infections studied.

R: We detailed this epidemiology (Please see new table 2; footnote).

5. Finally, could you elaborate on the criteria used to distinguish when CoNS are considered contaminants versus pathogens? Are CoNS included in the polymicrobial BSI counts, and if so, how is this determination made?

R: Thanks for the comment. The article specifically focuses on describing bacteremias considered significant. The definition of significant bacteremia has been clarified in the methods section (Methods section, lines 160-164, page 8).

Figure 1 and 2

1. Figure 1 could be considered converted into a stacked bar chart. This format would more clearly illustrate that *E. coli* and *Klebsiella* spp. are part of the overall total of microorganisms.

R: Done, see new Figure 1.

2. The time line in the graph should extend from 1995 to 2019, rather than from 1995 to 'overall.' Using a line graph to represent continuous time makes the inclusion of an 'overall' bar graphically inconsistent. Ending the line in 2019 would more effectively demonstrate the increasing trend in resistance, culminating in its highest level at the end of the observed period.

R: Done, see new Figure 1 and new figure 2.

3. Ensure that all abbreviations used in the figures are explained in their respective captions for clarity. This is particularly important for Figure 2, where readers need to understand the main antipseudomonal agents and the definitions of resistance patterns, such as MDR and XDR.

R: Done, see new Figure 2.

Table 3

Include numbers in the column headers for clarity. Additionally, ensure that the denominators for the percentages are clearly indicated, as it's currently not always apparent what they are based on.

R: Done, see new Table 3.

Discussion, line 184

Do you mean statistically significant or clinically significant? Consider using a more specific term to avoid ambiguity.

R: Done (please see discussion session, line 250, page 12).

Discussion, line 180-191

An important backdrop is whether your cancer population is representative of the broader cancer population in the city, region, or country? Or is the cancer population at this particular hospital a selected population with e.g., fewer breast cancers?

R: The Spanish healthcare system is a public health system, and there is no apparent

reason to think that our patients might differ from those in other centers in the city or country.

Discussion, line 200

A reference is needed to support the statement made here.

R: New reference has been added (Please see new ref. 22)

Discussion, line 203

Reconsider the use of the word "intrinsic"; perhaps "on its own" or a similar phrase would be more appropriate.

R: Done (please see Discussion section, line 271, page 13).

Discussion, line 209

Clarify whether "series" is intended to mean "studies."

R: We clarify this point (please see Discussion section, line 278, page 13).

Discussion, line 229-230

This appears to be an incomplete sentence.

R: We clarify this sentence (please see Discussion section, line 299-301, page 14).

Re: Spectrum02961-23R1 (Prevalence and Impact of Multidrug-Resistant Bacteria in Solid Cancer Patients with Bloodstream Infection: A 25-Year Trend Analysis.)

Dear Dr. Carolina García-Vidal:

Thank you for the privilege of reviewing your work. Below you will find my comments, instructions from the Spectrum editorial office, and the reviewer comments.

Thank you for your through attempts to address the reviewer comments.

A few remaining comments from myself

Major:

- I agree with the reviewer comment that analyzing trends with time as a continuous fashion (as I had previously mentioned)

"The authors mention that Chi square for trends was used to compare the different time spans. It seems to me that a more appropriate way to assess the statistical significance of The MDRO trends over time would be better analyzed by converting time(year) into a continuous variable and either performing a linear regression or Poisson/binomial over time with MDRO trends expressed as a rates."

I am referring to the trend analysis. I believe having time as a continuous variable (1 year increments) makes more sense than by 5 year increments as you lose the granularity offered by the long time period which is a strength of the study. I appreciate this has been done in prior publications by this group. However, I would be very appreciative if you could please provide further justification for this approach, or include in the next round of revisions. I am sorry if my prior comment was unclear..

Minor comments

-Please re-define abbreviations that have been defined in the abstract once more in the main text. For example SC is defined in the abstract (line 24), but needs to be defined once more in the body of the text. on line

-Please spell out species names the first time it is mentioned. For example, E. coli should be Escherichia coli the first time it is written, and can be E. coli after.

-142 : BSI should be BSI's

- Please provide a reference for the this statement "This exposure often leads to frequent contact with MDR-bacteria and extensive use of antibiotics that contributes to the selective pressure promoting the development of new resistant strains" (line 146)

-Line 185: Principio del formulario: unclear if this is a typo? Please translate to english

-line 209: you have repeated the mortality definition twice

-line 291 "in the last time span" : please include the years for this time span to aid readability

line 329: Please consider softening this statement. The use of the word mandatory is quite severe.

Revision Guidelines

Sincerely,
Ahmed Babiker
Editor
Microbiology Spectrum

Editor's comments:

Re: Spectrum02961-23R1 (Prevalence and Impact of Multidrug-Resistant Bacteria in Solid Cancer Patients with Bloodstream Infection: A 25-Year Trend Analysis.)

Dear Dr. Carolina García-Vidal:

Thank you for the privilege of reviewing your work. Below you will find my comments, instructions from the Spectrum editorial office, and the reviewer comments.

Thank you for your through attempts to address the reviewer comments.

A few remaining comments from myself

Major:

- I agree with the reviewer comment that analyzing trends with time as a continuous fashion (as I had previously mentioned). "The authors mention that Chi square for trends was used to compare the different time spans. It seems to me that a more appropriate way to assess the statistical significance of The MDRO trends over time would be better analyzed by converting time(year) into a continuous variable and either performing a linear regression or Poisson/binomial over time with MDRO trends expressed as a rates."

I am referring to the trend analysis. I believe having time as a continuous variable (1 year increments) makes more sense than by 5 year increments as you lose the granularity offered by the long time period which is a strength of the study. I appreciate this has been done in prior publications by this group. However, I would be very appreciative if you could please provide further justification for this approach, or include in the next round of revisions. I am sorry if my prior comment was unclear.

Response: We thoroughly appreciate the feedback and indeed find the statistical approach suggested to be highly relevant. Consequently, we are proceeding with a linear regression analysis to delineate with greater precision the temporal trend of multidrug resistance escalation. Please refer to methods section (page 7, lines 150-6 and page 8, lines 163-4), results section (page 9, lines 187-9) and new figure 1 for further details. Should the Editor wish for any additional analyses, please do not hesitate to inform us, as we recognize that the article stands to benefit from such enhancements. Thank you very much.

Minor comments

-Please re-define abbreviations that have been defined in the abstract once more in

the main text. For example SC is defined in the abstract (line 24) , but needs to be defined once more in the body of the text.

Done

-Please spell out species names the first time it is mentioned. For example, E. coli should be Escherichia coli the first time it is written, and can be E. coli after.

Done

-142 : BSI should be BSI's

Done

- Please provide a reference for this statement "This exposure often leads to frequent contact with MDR-bacteria and extensive use of antibiotics that contributes to the selective pressure promoting the development of new resistant strains" (line 146)

Done, please see references 8, 9 and 10.

-Line 185: Principio del formulario: unclear if this is a typo? Please translate to english
Sorry for the mistake. It's a typo and has been deleted.

-line 209: you have repeated the mortality definition twice

The repeated line has been deleted.

-line 291 "in the last time span" : please include the years for this time span to aid readability

Done.

- line 329: Please consider softening this statement. The use of the word mandatory is quite severe.

Done.

Re: Spectrum02961-23R2 (Prevalence and Impact of Multidrug-Resistant Bacteria in Solid Cancer Patients with Bloodstream Infection: A 25-Year Trend Analysis.)

Dear Dr. Carolina García-Vidal:

Your manuscript has been accepted, and I am forwarding it to the ASM production staff for publication. Your paper will first be checked to make sure all elements meet the technical requirements. ASM staff will contact you if anything needs to be revised before copyediting and production can begin. Otherwise, you will be notified when your proofs are ready to be viewed.

Sincerely,
Ahmed Babiker
Editor
Microbiology Spectrum